# SearchLVLMs: A Plug-and-Play Framework for Augmenting Large Vision-Language Models by Searching Up-to-Date Internet Knowledge

**Chuanhao Li**[2,1,3†], **Zhen Li**[2], **Chenchen Jing**[4], **Shuo Liu**[1], **Wenqi Shao**[1]
**Yuwei Wu**[2,3✉], **Ping Luo**[5,1], **Yu Qiao**[1], **Kaipeng Zhang**[1✉]

[1]OpenGVLab, Shanghai AI Laboratory [2]Beijing Institute of Technology
[3]Shenzhen MSU-BIT University [4]Zhejiang University [5]The University of Hong Kong

`https://nevermorelch.github.io/SearchLVLMs.github.io/`

## Abstract

Large vision-language models (LVLMs) are ignorant of the up-to-date knowledge, such as LLaVA series, because they cannot be updated frequently due to the large amount of resources required, and therefore fail in many cases. For example, if a LVLM was released on January 2024, and it wouldn't know the singer of the theme song for the new Detective Conan movie, which wasn't released until April 2024. To solve the problem, a promising solution motivated by retrieval-augmented generation (RAG) is to provide LVLMs with up-to-date knowledge via internet search during inference, *i.e.*, internet-augmented generation (IAG), which is already integrated in some closed-source commercial LVLMs such as GPT-4V. However, the specific mechanics underpinning them remain a mystery. In this paper, we propose a plug-and-play framework, for augmenting existing LVLMs in handling visual question answering (VQA) about up-to-date knowledge, dubbed SearchLVLMs. A hierarchical filtering model is trained to effectively and efficiently find the most helpful content from the websites returned by a search engine to prompt LVLMs with up-to-date knowledge. To train the model and evaluate our framework's performance, we propose a pipeline to automatically generate news-related VQA samples to construct a dataset, dubbed UDK-VQA. A multi-model voting mechanism is introduced to label the usefulness of website/content for VQA samples to construct the training set. Experimental results demonstrate the effectiveness of our framework, outperforming GPT-4o by ~30% in accuracy.

## 1 Introduction

Large vision-language models (LVLMs, *e.g.*, GPT-4V [19], Gemini Series [20], and Grok [21]) have received much attention for their impressive generative capabilities. They require a large resource for data collection, cleaning, and training, restricting them from frequently updating models. However, new information and knowledge are created every time, making LVLMs ineffective in many scenarios. For example, if we talk with LLaVA-1.6 [23] (released on January 30, 2024) about the new Detective Conan movie (realeased on April, 2024), such as "the singer of the theme song", it performs very badly. It is promising to augment LVLMs by retrieving up-to-date knowledge via internet search during inference, *i.e.*, internet-augmented generation (IAG). Although commercial LVLMs such as

---

[†] This work was done during the internship at Shanghai AI Laboratory.
[✉] Corresponding Authors: wuyuwei@bit.edu.cn; kp_zhang@foxmail.com

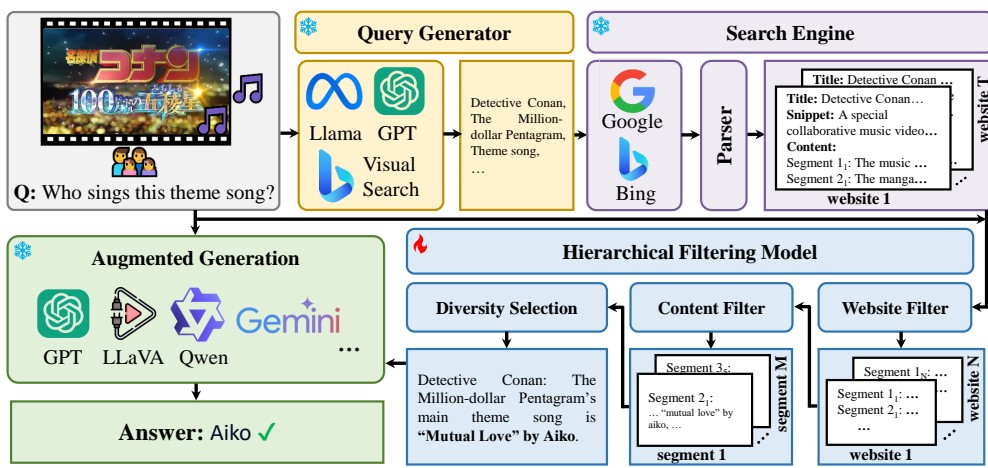

Figure 1: The proposed SearchLVLMs, a framework for LVLMs to access up-to-date knowledge.

GPT-4V [19] and Claude3 [22] have the ability of IAG, the specific mechanics underpinning them remain undisclosed. This paper proposes a plug-and-play framework to augment different LVLMs in handling visual question answering (VQA) about up-to-date knowledge, named SearchLVLMs.

We first introduce our overall framework applicable to different LVLMs for equipping them with up-to-date knowledge during inference. It consists of four components: query generator, search engine, hierarchical filtering model, and augmented generation, as shown in Figure 1. Specifically, we begin by extracting queries via Bing Visual Search and LLMs for an image-related question. Then, we acquire helpful websites through search engines and extract their contents by web scraping. However, it is impractical to augment LVLMs directly with the entire content of all websites, because: (1) Most LVLMs are poor at handling such long contexts. (2) Handling such long contexts is computationally intensive and time-consuming. To this end, a hierarchical filtering model is trained to find the most helpful content for answering the question, which first efficiently sifts the websites based on each website's title and snippet, and then identifies the most helpful content from the filtered websites. Finally, the filtered content is fed to LVLMs to assist them in answering the question.

We construct a dataset called UDK-VQA based on up-to-date news, used to train the hierarchical filtering model and evaluate our framework's performance. We propose a pipeline to automatically scrape the up-to-date news and generate news-related VQA samples. Specifically, we use search terms from Google Daily Search Trends and manually collected popular search terms as queries to search for hot news. For each news piece, we segment its content and ask GPT-3.5 to generate question-answer pairs from each segment. Then, we extract an entity for each question and replace it with its hypernym. To compose a VQA sample, we use Bing to search images of the replaced entity and cluster them to reduce the outliers among them. In doing so, answering the generated VQA samples requires models to consider both visual and textual information. We use queries from different time periods to scrape news from different time periods to generate samples for the training and test sets, avoiding test data exposure in the training data. In the training set, we further use a multi-model voting mechanism to label website's usefulness and content's usefulness for VQA samples, and combine the samples with websites and their content based on the label for training the hierarchical filtering model. In the testing set, we conduct manual screening to ensure its correctness.

To validate the effectiveness and generalizability of the proposed framework, we incorporate 15 state-of-the-art LVLMs into the framework, such as GPT-4V [19] and LLaVA-1.6 [23]. Notably, once the hierarchical filtering model is trained, our framework can adapt different LVLMs and improve their performance without any fine-tuning. Extensive experimental results demonstrate that our framework can significantly improve LVLM's ability to answer questions about up-to-date knowledge. Incorporating the LLaVA-1.6 model of our framework even outperforms the self-contained IAG-capable GPT-4o by ~30% in accuracy on UDK-VQA test set.

Our contributions are summarized as follows. (1) We propose the first open-source framework seamlessly incorporating existing LVLMs with up-to-date knowledge during inference. (2) We propose a pipeline that automatically generates VQA samples related to up-to-date news and construct

the first test set for evaluating LVLMs' ability to handle VQA on up-to-date knowledge. (3) Extensive experimental results on 15 state-of-the-art LVLMs demonstrate the effectiveness of our framework.

## 2 Related Work

### 2.1 Retrieval-Augmented Generation

Recently retrieval-augmented generation (RAG) attracted increasing attention of both the natural language processing [1, 9, 10, 2] and vision-and-language [3, 4, 18, 27, 28]. REALM [1] uses the query to retrieve the top $k$ most relevant article snippets, and uses large language models (LLMs) to generate $k$ responses, which are then combined to obtain a final output for question answering. Recently, [41, 10, 43] explores the internet-augmented generation (IAG) of LLMs to enable language models to access up-to-date information via search engines. Komeili *et.al.* [41] show that LLMs enhanced via search engines can generate less factually incorrect information during dialogue with humans. Lazaridou *et.al.* [10] uses few-shot prompting to enable LLMs to exploit knowledge returned from Google search to answer questions about factual and up-to-date information. In vision-and-language, REVEAL [3] builds a memory by encoding open-world knowledge including image-text pairs, question-answering pairs, etc., and uses a retriever to find the most relevant knowledge entries in the memory. The memory, encoder, retriever, and generator are pre-trained in an end-to-end manner. Re-ViLM [4] augments Flamingo [5], by retrieving relevant image-text pairs from the external image-text datasets [6, 7, 8] for zero and in-context few-shot image-to-text generations. RA-CM3 [42] performs retrieval from an external memory for generating images and text. Differently, we focus on enabling LVLMs to retrieve up-to-date knowledge via Internet search during inference.

### 2.2 Large Models with Search Engine

Recent years have witnessed a growing interest in exploring external tools for LLMs [11, 15, 12, 16, 13, 52]. Among them, some methods [13, 14, 17] can use search engines to access up-to-date knowledge. Nonetheless, these methods usually focus on how to appropriately use different tools to enhance LLMs, such as using Python interpreter to generate complex programs [15], incorporating more external tools [13], or updating tools by acquiring new knowledge [17]. Although they can access up-to-date knowledge, they usually directly use the website snippets for augmenting generation. By contrast, this work focuses on internet-augmented generation and explores how to obtain more relevant up-to-date knowledge and effectively use retrieved knowledge to augment LVLMs.

## 3 SearchLVLMs Framework

In this section, we introduce SearchLVLMs, a framework that seamlessly incorporate existing LVLMs, allowing these LVLMs to access up-to-date knowledge without fine-tuning. The whole framework is illustrated in Figure 1. For a natural language question $Q$ about an image $V$, we first extract queries for both $Q$ and $V$ via the query generator. Then we enter the queries into search engines, and the search engine would return related websites, each of which consist of a title and a snippet. To identify the most helpful content within the websites, a website filter is used to filter the websites based on their titles and snippets, and a content filter is further used to filter the content of the websites filtered by the website filter. Finally, we stitch the filtered content together to prompt existing LVLMs.

### 3.1 Query Generator

**Question Query Generator.** To get queries that make search engines return websites containing helpful content, we leverage large language models (LLMs) to extract queries for $Q$. Thanks to the language understanding capability of LLMs, the role played by each word can be well inferred from the grammatical information of $Q$, even if certain words are unknown for the LLMs. We use "*Do not try to answer the question, just print the most informative no more than three entities in the question. Put them on one line and separate them with comm.*" to prompt LLMs to generate queries.

**Image Query Generator.** For an image $V$, we leverage Bing Visual Search to analyze the image entities of $V$ as queries. The reason for using Bing Visual Search rather than a LVLM to extract queries for $V$ is that current LVLMs are inadequate in extracting image entities especially for emerging

entities. Notably, Bing Visual Search is a tool different from commonly used search engines, returning image-related attributes, including image entity names, image-related search terms and image-related websites. However, entity names are missing in most cases. To address this problem, we extract the longest public ancestor of related search terms and related website titles as the queries for $V$.

## 3.2 Search Engine

The extracted queries are fed into a search engine, and the search engine returns relevant websites with their titles and snippets. However, the returned titles and snippets often contain limited and incomplete information. For example, for a website with title "*Pororo Dragon Castle Adventure*", the entire snippet returned by Bing is "*Pororo and his friends were having fun when a little red dragon named Arthur appears above them Arthur who claims to be the king of dragons commands Pororo and his friends to search for his Dragon ...*", obviously there is more about "*Pororo Dragon Castle Adventure*" contained in the website. Thus we parse the textual content of all websites. For a website, not all of its content contributes to answering questions, we empirically divide the website content into segments every third sentence for a more granular selection of content.

## 3.3 Hierarchical Filtering Model

Since most of the existing LVLMs cannot receive long context as inputs, and long contexts can be computationally intensive and time consuming for them, it's necessary to filter the website content after obtaining the websites via the search engine. Towards this goal, we train a hierarchical filtering model, which consists of a website filter and a content filter to perform a two-step filtering.

**Website Filter.** The aim of the website filter is to perform the filtering of websites based on their titles and snippets. Specially, a website scoring model is trained via instruction tuning, to predict how helpful a website will be in answering a question, and the $N$ websites with higher scores would be kept. The training samples are in the format $(T, S, Q, V, R_w)$, where $R_w$ is a quantitative usefulness in the interval $[0, 1]$ representing how helpful a website with title $T$ and snippet $S$ will be in answering $Q$ related to $V$. Based on the samples, we construct instructions like "*How helpful is an article with such a title and snippet in answering the question based on the image? Choose the best option. Title: <T> Snippet:  Question: <Q> Options: A. 1.0 B. 0.8 C. 0.6 D. 0.4 E. 0.2 F. 0.0*". In doing so, the score regression problem is converted into a classification problem, which is easier to learn.

**Content Filter.** The content filter is used to select the most helpful content segments from the websites filtered by the website filter. For each content segment, we predict how helpful is it for answering $Q$ by a content scoring model. The content scoring model is trained by samples in the format $(C, Q, V, R_c)$, where $C$ is a content segment, and $R_c$ is the quantitative usefulness of $C$ in answering $Q$. The instructions for training the content scoring model are in the format: "*How helpful is this context in answering the question based on the image? Choose the best option. Context: <C> Question: <Q> Options: A. 1.0 B. 0.8 C. 0.6 D. 0.4 E. 0.2 F. 0.0*". We use the model to sort all content segments and select the $M$ highest scoring ones as the obtained segments.

**Diversity Selection.** To avoid LVLMs answer questions using bias from repetitive contexts, we performed a quadratic selection on the obtained segments based on diversity. Specially, we extract CLIP features [37] for all the segments and cluster them using k-means [44]. The segments closest to the center of each cluster are stitched together as the final obtained content for prompting the LVLMs.

## 3.4 Augmented Generation

We augment existing LVLMs by prompting them with the final obtained content. Taking answer the multiple choice questions as an example, for a question $Q$ with candidate answers $A_1$, $A_2$, $A_3$ and $A_4$, we use the prompt "*Given context: <X> Question: <Q> Answers: A.<A_1> B.<A_2> C.<A_3> D.<A_4> Answer with the option's letter from the given choices directly based on the context and the image.*", where $X$ denotes the final content obtained by the hierarchical filtering model.

# 4 UDK-VQA Dataset

To evaluate the effectiveness of our framework, we propose a pipeline to automatically scrape the up-to-date news and generate news-related VQA. The whole pipeline is demonstrated in Figure 2.

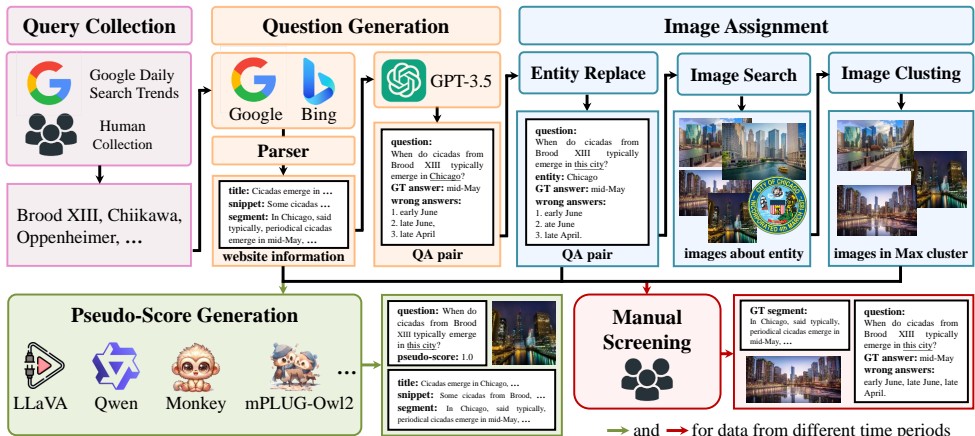

Figure 2: Overall pipeline of the sample generation for the UDK-VQA dataset. For brevity, we only show one output item at several steps, such as the content segment returned by the Parser. Notably, we use queries from different time periods to scrape news from different time periods to generate training samples and test samples, which is not reflected in this figure for brevity.

The pipeline is also used to collect training samples for the hierarchical filtering model. We first collect hot search terms as queries to scrape relevant news returned by search engines. For each piece of news, every third sentence is divided into a segment. We then employ GPT 3.5 to generate a question-answer pair for each segment, and extract an entity in the question, replacing it with its hypernym. Bing Image Search is used to find images for the replaced entity, and after removing outliers from the images using clustering, the images and the question-answer pair are composed into VQA samples. We combine the VQA samples and website information (*e.g.*, title, snippet and content), and introduce a multi-model voting mechanism to generate pseudo-score, constituting the training set. For the test set, manual screening is conducted to ensure the correctness of test samples.

## 4.1 Query Collection

Google daily search trends is an available data source that reflects what's hot in real time, and is well suited as the query used to construct our dataset. However, we observe that most search terms of the Google daily search trends are related to politics and sports, which poses a great limitation. Therefore, we further manually collect popular search terms to improve the query diversity. The popular search terms are collected from many other domains including films, technological products, anime characters, places of interest, and so on. These human-collected queries were mixed with queries from Google daily search trends to be used for subsequent sample generation.

## 4.2 Question Generation

For each query, we use Bing to search for relevant and up-to-date news. We divide every third sentence int the scraped news content into a segment, and prompt GPT-3.5 to generate a question-answer pair and several confused answers for each segment by: "*Given context: <Con> Filling the blanks to generate a question about the most informative event of the context, generate an correct answer to the question in no more than three words based on context, and generate three incorrectly confused answers of no more than three words based on context. Question: ___ Correct answer: ___ Incorrect answers: A. ___ B. ___ C. ___*", where *<Con>* denotes a segment. We design a simple but effective rule to ensure the correctness of the generated pairs, which requires a model can answer a question $Q$ with $A$ based on a segment $C$, if the model can generate a question-answer pair $(Q, A)$ based on $C$.

## 4.3 Image Assignment

To generate VQA samples and avoid the model's reliance on language priors for answering, we create samples that necessitate an understanding of the image for correct answers. Firstly, we extract an entity for each question via named an entity recognition (NER) model [38]. Images of the entity

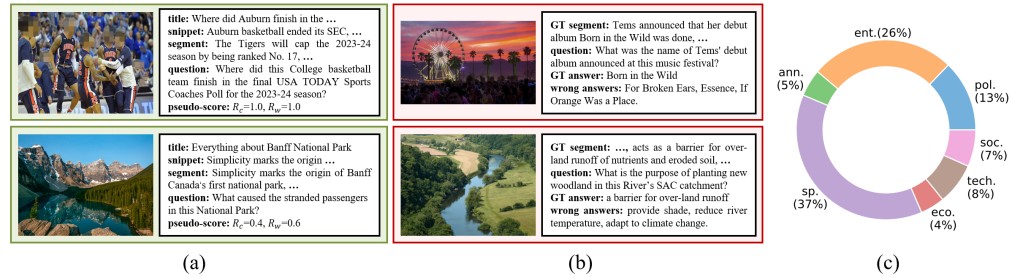

Figure 3: (a) Training samples. (b) Test samples. (c) Category statistics for the test set of UDK-VQA.

are then obtained by Bing Image Search. Since the images returned by the search engine are noisy (outlier images), we cluster them based on the CLIP feature [37] of the images and keep only the images in the cluster with the highest number of images. Finally, the kept images are assigned to the new question-answer pairs where the entity is replaced by its hypernym, to compose VQA samples. An obtained VQA sample can be denoted as $(V, Q, A_{gt}, \{A_w^i\}_{i=1}^3)$, where $Q$ is the generated question, $V$ is the entity image, $A$ represents the ground-truth answer, and $A_w^i$ is $i$-th confused wrong answer.

## 4.4 Pseudo-Score Generation

For a VQA sample generated from the content segment $C$, which we denote as the ground-truth segment for the sample, it is certain that $C$ is most helpful in answering this sample. Inevitably, we must consider to what extent do the other content segments contribute to answering the sample? We propose a pseudo-score generation method that uses five LVLMs for voting to quantify how helpful a content segment is to a VQA sample into six values: 1.0, 0.8, 0.6, 0.4, 0.2 and 0.0. Specially, for a VQA sample with the ground-truth segment $C$ from a news fetched for a query, we first sample four content segments from the news for the query beyond its ground-truth segment. Then we use each sampled segment to prompt each of the five LVLMs to answer the VQA sample and count the rate of LVLMs that answer correctly as the pseudo-score for the segment.

In doing so, we obtain training samples for the content filter, in the format $(C, Q, V, R_c)$, where $C$ is a content segment, $Q$ denotes a question related to the image $V$, and $R_c$ is the pseudo-score of how helpful $C$ is to answer $Q$. Moreover, we count the maximum pseudo-score of all content segments in a news for a VQA sample as the pseudo-score for the news website, dubbed $R_w$, to build training samples for the website filter. The training sample format for the website filter is $(T, S, Q, V, R_w)$, where $T$ is the website title, $S$ is the website snippet. By merging these training samples into the training instructions mentioned in Section 3.3, the hierarchical filtering model can be implemented.

## 4.5 Manual Screening

For constructing the test set, we do not use the pseudo-score generation method. A test sample $(C, V, Q, A_{gt}, \{A_w^i\}_{i=1}^3)$ can be seen as a VQA sample with its ground-truth content segment $C$. It is worth noting that $C$ is only provided when testing the upper bound of performance. For each test sample, we randomly mix $A_{gt}$ and $\{A_w^i\}_{i=1}^3$, then assign them the options (*i.e.* A, B, C and D), and add a complementary option *E. No Correct Answers*, to evaluate LVLMs in a multiple choice format. Moreover, we manually review all test samples to ensure that they are correct.

## 4.6 Dataset Analysis

To prevent test data exposure in the training set, we use queries and news from different time periods to construct the training and test sets. For the training set, we use queries from February 17 to March 31, 2024, to scrape news before April 10, 2024. The training sample counts for the website and content filters are 599,700 and 850,267. For the test set, queries from April 1 to April 31, 2024, are used to scrape news after April 10, yielding 1,000 test samples. We divide the test sample into seven categories: politics, entertainment, announcement, sports, economic, technology and society, based on their required knowledge. We visualize some samples and the statistics for test samples in Figure 3. A new version, UDK-VQA-20240925, is constructed, with details in the **Appendix**.

# 5 Experiments

## 5.1 Settings

**Training.** We implement two versions of the hierarchical filtering model, one using LLaVA-1.5-vicuna-7b [39] and the other using Qwen-VL-Chat [29]. In each version, we use same hyper-parameters to fine-tune two same LVLMs with LoRA [24] as the website filter and the content filter, respectively. Whether fine-tuning LLaVA-1.5-vicuna-7b or Qwen-VL-Chat, the entire training process is facilitated on two Nvidia A100 GPUs, using a batch size of 128 over 3 epochs.

**Baselines.** We incorporate 15 representative LVLMs into the proposed framework including Gemini 1.5 Pro [20], GPT-4V [19], GPT-4o, InternVL-1.5 [45], LLaVA-1.6 [23], LLaVA-1.5 [39] XComposer2 [26], Monkey [25], CogVLM [30], MiniCPM-V2 [35], mPLUG-Owl2 [31], Qwen-VL [29], MMAlaya [34], Xtuner [32] and VisualGLM [33]. We implement Gemini 1.5 Pro, GPT-4V and GPT-4o via their official webs and APIs. We implement other LVLMs based on VLMEvalKit [36].

**Evaluation.** We evaluate LVLMs on four datasets including GQA [46], InfoSeek [47], A-OKVQA [48] and the proposed UDK-VQA. The reason behind selecting GQA, InfoSeek, and A-OKVQA is to evaluate the generalization capability of our framework across datasets that do not necessitate up-to-date knowledge. In addition to evaluating LVLMs via VLMEvalKit, we design additional matching patterns for each LVLM with respect to its answer format. For example, we additionally use the pattern "The answer is XXX." for XComposer2 as it often answers in this format. All evaluations are conducted with a single Nvidia A100 GPU.

## 5.2 Quantitative Comparison with SOTA LVLMs

We compare with state-of-the-art LVLMs on the UDK-VQA test set, including Gemini 1.5 Pro [20], GPT-4V [19], GPT-4o, LLaVA-1.6 [23] and InternVL-1.5 [45]. For Gemini 1.5 Pro, GPT-4V and GPT-4o, we implement their *Raw* version via official APIs, which do not have the ability of IAG.

Table 1: Comparision with SOTA LVLMs on UDK-VQA, where "Raw" represents the model without IAG ability (*e.g.*, official API version), "IAG" represents the model with self-contained IAG-capable ability (official web version), "LC" represents the model with long context input. "Gen.", "Cham." and "CLIP→FID (C→F)" denote the method from [51], [13] and [47], respectively. "★" indicates that the method leverages our framework to access up-to-date knowledge. "Ours" stands for incorporating the Raw baseline into our framework. The value outside/in () indicates the accuracy over samples that do not violate the content management policy of current/all model(s).

| Model | Variant | pol. | ent. | ann. | sp. | eco. | tech. | soc. | overall |
|---|---|---|---|---|---|---|---|---|---|
| Gemini 1.5 Pro | Raw | 6.2 (5.7) | 15.8 (16.3) | 10.2 (11.9) | 7.4 (8.1) | 2.3 (2.5) | 8.0 (6.2) | 3.0 (3.7) | 9.1 (9.5) |
| | LC | 61.7 (65.7) | 71.5 (77.3) | 73.5 (76.2) | 77.2 (79.2) | 72.7 (72.5) | 81.3 (83.1) | 62.1 (66.7) | 76.4 (76.1) |
| | **Ours** | **82.8 (82.9)** | **79.6 (79.0)** | **91.8 (92.9)** | **81.5 (80.1)** | **97.7 (97.5)** | **84.0 (83.1)** | **90.9 (88.9)** | **83.3 (82.3)** |
| GPT 4V | Raw | 21.1 (23.8) | 31.5 (30.9) | 16.3 (19.0) | 16.7 (17.5) | 15.9 (17.5) | 41.3 (38.5) | 21.2 (22.2) | 24.2 (23.8) |
| | IAG | 62.5 (68.0) | 61.9 (63.6) | 63.3 (66.7) | 62.4 (63.3) | 70.5 (67.5) | 80.0 (78.5) | 69.7 (70.4) | 64.5 (65.9) |
| | **Ours** | **76.6 (83.8)** | **85.8 (85.8)** | **89.8 (90.5)** | **86.2 (86.4)** | **97.7 (97.5)** | **92.0 (90.8)** | **87.9 (92.6)** | **87.2 (87.4)** |
| GPT 4o | Raw | 36.7 (40.0) | 34.2 (36.1) | 42.9 (47.6) | 28.6 (30.4) | 40.9 (42.5) | 65.3 (67.7) | 36.4 (38.9) | 37.2 (37.8) |
| | IAG | 61.7 (63.1) | 57.3 (58.4) | 69.4 (64.3) | 48.4 (47.9) | 70.5 (75.0) | 81.3 (81.5) | 62.1 (61.1) | 57.8 (57.9) |
| | **Ours** | **86.7 (92.4)** | **89.6 (91.4)** | **98.0 (100)** | **83.9 (88.0)** | **97.7 (100)** | **90.7 (96.9)** | **89.4 (94.4)** | **91.8 (91.6)** |
| LLaVA 1.6 | Raw | 43.8 (45.7) | 32.3 (31.8) | 22.4 (23.8) | 24.9 (23.2) | 25.0 (25.0) | 53.3 (55.4) | 33.3 (31.5) | 31.8 (31.2) |
| | Gen. | 39.1 (40.0) | 31.5 (28.8) | 18.4 (19.0) | 25.7 (25.3) | 36.4 (37.5) | 44.0 (44.6) | 39.4 (37.0) | 31.3 (30.4) |
| | Cham. | 58.6 (58.1) | 57.3 (57.5) | 53.1 (57.1) | 65.3 (67.2) | 52.3 (52.5) | 72.0 (67.7) | 74.2 (72.2) | 62.3 (62.7) |
| | C→F★ | 55.5 (56.2) | 56.5 (57.9) | 34.7 (35.7) | 54.0 (54.5) | 54.5 (52.5) | 62.7 (64.6) | 56.1 (53.7) | 54.7 (55.3) |
| | **Ours** | **86.7 (87.6)** | **91.9 (91.8)** | **93.9 (97.6)** | **88.1 (88.0)** | **90.9 (90.0)** | **93.3 (93.8)** | **95.5 (100)** | **90.2 (90.7)** |
| Intern VL 1.5 | Raw | 43.8 (43.8) | 53.1 (52.4) | 49.0 (47.6) | 29.9 (30.7) | 34.1 (35.0) | 73.3 (76.9) | 37.9 (35.2) | 42.6 (42.8) |
| | Gen. | 29.7 (30.5) | 28.1 (26.2) | 28.6 (26.2) | 22.8 (23.5) | 31.8 (32.5) | 42.7 (46.2) | 28.8 (27.8) | 27.6 (27.6) |
| | Cham. | 59.4 (58.1) | 61.9 (61.8) | 55.1 (57.1) | 55.6 (55.4) | 52.3 (52.5) | 65.3 (67.7) | 71.2 (70.4) | 59.3 (59.2) |
| | C→F★ | 59.4 (58.1) | 65.0 (64.8) | 44.9 (42.9) | 54.2 (55.7) | 47.7 (50.0) | 65.3 (66.2) | 53.0 (50.0) | 57.7 (58.0) |
| | **Ours** | **90.6 (89.5)** | **95.4 (95.3)** | **98.0 (97.6)** | **88.9 (88.0)** | **100 (100)** | **96.0 (95.4)** | **98.5 (98.1)** | **92.9 (92.3)** |

Table 2: Experiments on GQA [46], InfoSeek [47], A-OKVQA [48], where GQA does Not Rely on external Knowledge (NRK), InfoSeek and A-OKVQA Rely on Commonsense Knowledge (RCK).

| Model | Variant | Retrieval Sources | | Datasets | | |
|---|---|---|---|---|---|---|
| | | Local Data (clear GT) | Internet Data (unclear GT) | GQA (NRK) | InfoSeek (RCK) | A-OKVQA (RCK) |
| CFR [50] | - | - | - | 72.10 | - | - |
| Oracle→FID [47] | - | ✓ | - | - | 45.60 | - |
| Omni-SMoLA [49] | - | ✓ | - | - | - | 84.10 |
| LLaVA-1.6 | Raw | - | - | 61.66 | 37.86 | 75.53 |
| | **Ours** | - | ✓ | **62.33** | **41.25** | **76.22** |
| InternVL-1.5 | Raw | - | - | 74.03 | 51.13 | 84.53 |
| | **Ours** | - | ✓ | **74.41** | **53.10** | **84.59** |

Since Gemini 1.5 Pro is famous for receiving long contexts, we use all website content returned by the search engine of our framework to prompt it directly, dubbed **LC**. For GPT-4V and GPT-4o, we test their self-contained IAG-capable ability via prompting their official web versions with "*Retrieve relevant news and answer the question directly from the given options using the option letters based on the image.*", dubbed **IAG**. We incorporate each Raw baseline into our framework as **Ours**.

The experimental results on UDK-VQA are listed in Table 6, we can observe that: (1) InternVL-1.5 with our framework achieves the best performance on almost categories of UDK-VQA. (2) For all four baselines, our framework consistently improves their accuracy (e.g., 22.7% and 34.0% absolute performance gains in overall accuracy for GPT-4V and GPT-4o, respectively). (3) Our framework uses shorter contexts but has higher accuracy (e.g., 76.4% vs 83.3% in accuracy for LC and Ours variants of Gemini, respectively). The observations suggest that our framework is generalizable and effective in enhancing the ability of LVLMs to answer questions about up-to-date knowledge.

In addition, the experimental results on GQA, InfoSeek and A-OKVQA are listed in Table 2. Since these datasets do not rely on the up-to-date knowledge, we use a simple strategy to avoid misleading LVLMs with the up-to-date knowledge by invoking our framework when they respond with "E" (as mentioned in Section 4.5) without retrieval. From the table, we can observe that our framework improves the performance of different LVLMs across various datasets. The improvements on these three datasets are not as significant as on our UDK-VQA dataset for the following reasons: (1) The GQA dataset does not rely on external knowledge and is used to evaluate the reasoning ability of LVLMs, which is beyond the scope of our framework. (2) Our framework focuses on retrieving the up-to-date knowledge, whereas the InfoSeek dataset and the A-OKVQA dataset rely on commonsense knowledge, much of which has already been used in the training data of LVLMs.

## 5.3 Ablation Studies

The experimental results of ablation studies on the proposed UDK-VQA dataset are shown in Table 3, where we use LLaVA-1.6 [23] as the baseline. Firstly, we investigate simple IAG methods, including using the similarity between questions and segments to select segments, *i.e.*, **IAG (SIM Q)**, using the similarity between images and segments to select segments *i.e.*, **IAG (SIM V)**, using the averaged similarity of the the above two similarities to select segments, *i.e.*, **IAG (SIM QV)**. These methods show limited improvements and achieve unsatisfactory accuracy.

Then, we study the influences of different components of our framework on the performance. For the hierarchical filtering model, we study two popular LVLMs, LLaVA-1.5 [39] and Qwen-VL [29]. For the query generator, we conduct experiments with NER [38], LLaMA3 [40], GPT-3.5 and Bing Visual Search. We observe that: (1) Using different backbone for the hierarchical filtering model has little effect on performance. (2) Using multiple question query generators at the same time can result in better performance than using only one. (3) Using both the question query generator and the image query generator gives the best performance. These observations suggest that all components of our framework are effective in improving the baseline, and components are complementary to each other.

Table 3: Ablation studies of our framework on UDK-VQA.

| Model | Variant | Hierarchical Filtering Model | | Query Generator (Q) | | | Query Generator (V) | Acc. |
|---|---|---|---|---|---|---|---|---|
| | | *LLaVA-1.5* | *QWen-VL* | *NER* | *LLaMA3* | *GPT-3.5* | *Bing Visual Search* | (%) |
| LLaVA-1.6 | Raw | - | - | - | - | - | - | 31.8 |
| | IAG (SIM $Q$) | - | - | - | - | - | - | 46.1 |
| | IAG (SIM $V$) | - | - | - | - | - | - | 47.1 |
| | IAG (SIM $QV$) | - | - | - | - | - | - | 47.7 |
| | **Ours** | ✓ | - | - | - | - | ✓ | 49.3 |
| | | ✓ | - | - | - | ✓ | - | 65.9 |
| | | ✓ | - | ✓ | - | - | ✓ | 81.4 |
| | | ✓ | - | - | ✓ | - | ✓ | 86.6 |
| | | ✓ | - | - | - | ✓ | ✓ | 87.6 |
| | | ✓ | - | ✓ | ✓ | ✓ | ✓ | **90.2** |
| | | - | ✓ | ✓ | ✓ | ✓ | ✓ | 89.6 |

## 5.4 Analysis of Pseudo-Score Generation

We analyze the influences of using different LVLMs to generate pseudo-scores on the performance. We categorize 10 LVLMs into two groups based on their released date, the first group contains LLaVA-1.6, XComposer2, Monkey, CogVLM and MiniCPM-V2, the second group contains mPLUG-Owl2, Qwen-VL, MMAlaya, Xtuner and VisualGLM. Using these two groups to generate pseudo-scores are dubbed ***PSG with G1*** and ***PSG with G2***. Experimental results are shown in Figure 4, which reveal that: (1) The proposed framework can be directly used to boost LVLMs that are not used for generating pseudo-scores, which show the transferability of our framework. (2) The use of more recent LVLMs for generating pseudo-scores allows for greater improvements in general. (3) Different LVLMs have different performance upper bound, some of them achieve limited accuracy (*e.g.*, $\sim 70\%$ in accuracy for VisualGLM) even are augmented with ground-truth segments (***GT Segment***).

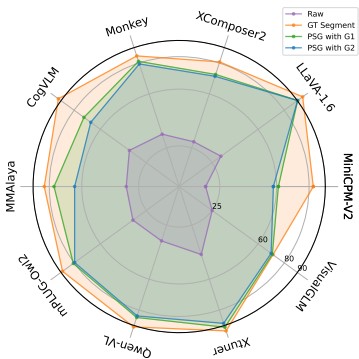

Figure 4: Accuracy using different LVLMs to generate pseudo-scores.

## 5.5 Analysis of Training Strategy

Would jointly training the hierarchical filtering model with LVLMs result in greater improvements? We use LLaVA-1.5 [39] as the backbone for the hierarchical filtering model and conduct experiments with Qwen-VL [29] and LLaVA-1.5 as the LVLMs. As shown Table 4, separate training, where the LVLMs are frozen during the training of the hierarchical filtering model, leads to a more significant improvement in performance. The main reasons are: (1) Our training data uses pseudo-labeling instead of high-quality human annotation. Training based on such data may cause LVLMs to lose their original semantic understanding capabilities. (2) Our training and testing sets are generated from news from different time periods, involving different entities and having different distributions. Training LVLMs on our training set easily leads to overfitting, resulting in lower generalization on the test set.

Table 4: Experiments of different training strategies on UDK-VQA.

| LVLM | Variant | Training Strategy | Acc (%) |
|---|---|---|---|
| QWen-VL | Raw | - | 35.2 |
| | Ours | Joint | 68.5 |
| | **Ours** | **Separate** | **84.8** |
| LLaVA 1.5 | Raw | - | 41.2 |
| | Ours | Joint | 68.0 |
| | **Ours** | **Separate** | **88.9** |

## 5.6 Analysis of Diversity Selection

In this section, we investigate the necessity of diversity selection. We compare our diversity selection (Div-$K$) with Top-$K$ selection, and the experimental results of 10 LVLMs are shown in Figure 5.

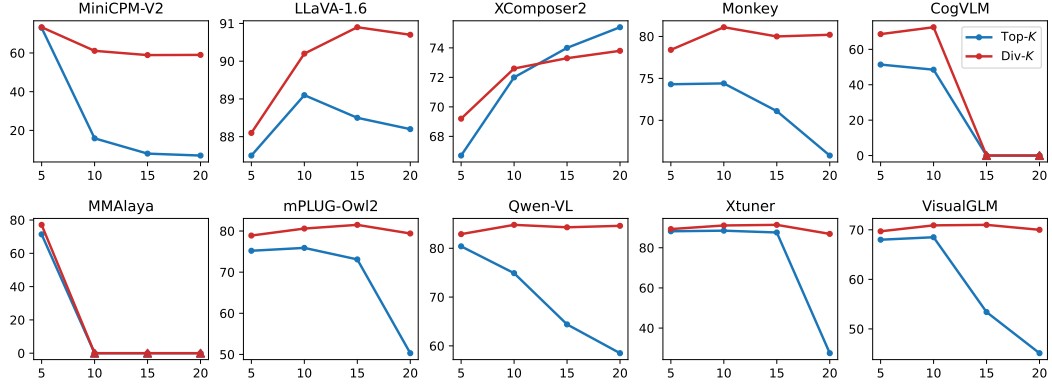

Figure 5: Comparison between Top-$K$ selection and diversity selection (Div-$K$), where $K$ denotes the number of stitched content segments for prompting LVLMs. For each sub-figure, the horizontal coordinate is $K$ and the vertical coordinate is the accuracy. Note that an accuracy of 0 means that the model fails at the context length under the current setting of $K$, and is labeled as a triangle.

The Top-$K$ selection means stitching $K$ content segments with the highest scores together to prompt the LVLMs. For Div-$K$, $K$ denotes the number of clusters. Experimental results demonstrate that: (1) Our diversity selection outperforms the Top-$K$ selection regardless of the setting of $K$ for most LVLMs. (2) As $K$ increases, the performance using the Top-$K$ selection plummets. This is because content with high scores is similar, and if a LVLM receives too many duplicate content as inputs, it will misinterpret the instruction and thus repeat the inputs instead of answering the question. These experimental results prove the necessity and effectiveness of the diversity selection.

## 5.7 Analysis of Website Filter

An important capability of the website filter is the trade-off between the content filter efficiency and the LVLMs' accuracy. Adjusting the filtered website number $N$ can control the token number that the content filter needs to process as a percentage of the total token number returned by the search engine, dubbed $\theta$. The variation in accuracy of LVLMs as $\theta$ increases is shown in Figure 6, we can observe that: (1) The accuracy of LVLMs increases with $\theta$, especially when $\theta \leq 40\%$. (2) The increase in accuracy of LVLMs slows down after $\theta \geq 40\%$. Therefore, setting $\theta = 40\%$ achieves a better trade-off, because the accuracy obtained by processing 40% tokens is close to 98% of the accuracy obtained when processing 100% tokens.

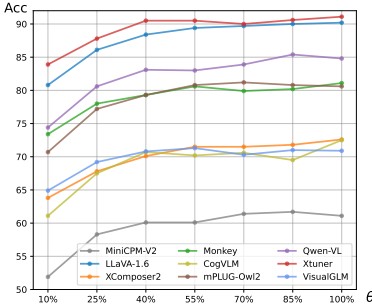

Figure 6: Accuracy under the content filter processing different percentages of website content.

## 5.8 Analysis of Snippet Completeness

We consider whether providing complete snippets to the website filter could result in further improvements. The experimental results of snippet completeness are presented in the **Appendix**.

## 6 Conclusion

In this work, we have presented SearchLVLMs, a plug-and-play framework to augment LVLMs in handling visual question answering about up-to-date knowledge. By introducing a hierarchical filtering model, the framework enables LVLMs to access up-to-date knowledge. A UDK-VQA dataset is further curated by scraping up-to-date news and generating news-related VQA samples. The dataset enables quantitatively evaluate the ability of LVLMs to respond to questions about up-to-date knowledge. Experimental results on UDK-VQA demonstrate that our framework can significantly boost the performance of LVLMs for answering questions requiring up-to-date knowledge.

**Acknowledgments** This work was supported by the Natural Science Foundation of China (NSFC) under Grants No. 62172041 and No. 62176021, Natural Science Foundation of Shenzhen under Grant No. JCYJ20230807142703006, Key Research Platforms and Projects of the Guangdong Provincial Department of Education under Grant No.2023ZDZX1034, and the China Postdoctoral Science Foundation (No. 2023M743003). This work was partially supported by the National Key R&D Program of China (No.2022ZD0161000, No.2022ZD0160101, No.2022ZD0160102) and the General Research Fund of Hong Kong No.17200622 and 17209324.

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

# A  Appendix

## A.1  Analysis of Snippet Completeness on UDK-VQA

The incomplete snippets returned by search engines lead us to consider whether providing complete snippets to the website filter could result in further improvements. We present the experimental results of snippet completeness on our UDK-VQA dataset in the Table 5, where $\theta$ represents the percentage as mentioned in Section 5.7. For each website snippet, we attempt to locate the full sentence corresponding to the snippet by crawling the website's content. However, the content of many websites could not be crawled. For such websites, we experiment with two strategies: (1) Discarding these websites during training and testing. (2) Using the incomplete snippets. The experimental results are shown in the table below, where "***Raw***" represents all snippets without completion, "***Discard***" represents strategy (1), and "***Mixture***" represents strategy (2). From the experimental results we can observe that directly discarding the websites leads to a significant performance loss, as discarding reduces the number of usable websites by approximately half, thereby limiting the performance of the website filter. Furthermore, as $\theta$ increases, the performance of strategy (2) becomes increasingly close to that of all snippets without completion (*i.e.*, Raw), which validates that the completeness of the snippets has little impact on accuracy.

Table 5: Experiments of snippet completeness.

| Baseline | Variant | $\theta$ | | | | | | |
|---|---|---|---|---|---|---|---|---|
| | | 10% | 25% | 40% | 55% | 70% | 85% | 100% |
| LLaVA-1.6 (Ours) | Raw | 80.8 | 86.1 | 88.4 | 89.4 | 89.7 | 90.0 | 90.2 |
| | Discard | 76.6 | 80.2 | 80.7 | 82.1 | 81.9 | 81.3 | 81.6 |
| | Mixture | **83.6** | **87.9** | **89.4** | **89.6** | **89.8** | **90.4** | **90.2** |
| InternVL-1.5 (Ours) | Raw | 84.6 | 89.2 | 91.1 | **92.9** | **92.4** | **92.7** | 92.9 |
| | Discard | 79.7 | 82.2 | 82.9 | 83.8 | 84.0 | 82.8 | 83.2 |
| | Mixture | **86.0** | **89.5** | **92.2** | 92.3 | 92.2 | 92.5 | **92.9** |

## A.2  UDK-VQA-20240905 Dataset

To test the capabilities of LVLMs to handle visual question answering on the latest up-to-date knowledge, we construct a new UDK-VQA-20240905 test set based on the proposed data generation pipeline. For the UDK-VQA-20240905 test set, we use the queries of Google Daily Search Trends from September 1, 2024 to September 5, 2024, to scrape news after September 5, 2024. In addition, we manually construct some test samples to ensure that the UDK-VQA-20240905 test set covers a wider range of categories of knowledge. The total number of test samples is 50, and the samples can be divided into five categories, including game (5), sports (24), society (8), entertainment (9), and economic (4), based on their required knowledge. The number in parentheses after each category indicates the number of samples in that category. As shown in Figure 7, answering the samples of UDK-VQA-20240905 requires up-to-date knowledge. For example, the first sample with the question "Where is the Meditation Spot located at this Temple?" is about the game "Black Myth: Wukong," which was released on August 20, 2024, after the release date of existing LVLMs.

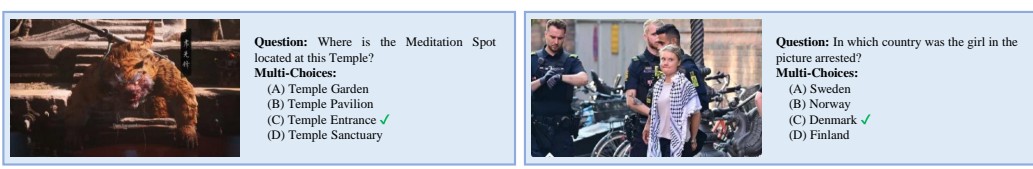

Figure 7: Test samples of the UDK-VQA-20240905 test set.

Table 6: Comparison with state-of-the-art LVLMs on UDK-VQA-20240905.

| Model | Variant | game | sp. | soc. | ent. | eco. | overall |
|---|---|---|---|---|---|---|---|
| Gemini 1.5 Pro | Raw | 0.0 | 0.0 | 0.0 | 11.1 | 0.0 | 2.0 |
| | **Ours** | **20.0** | **58.3** | **75.0** | **55.6** | **100** | **60.0** |
| GPT-4o | Raw | 20.0 | 25.0 | 12.5 | 22.2 | 50.0 | 24.0 |
| | IAG | 40.0 | 70.8 | 62.5 | 55.6 | 100 | 66.0 |
| | **Ours** | **80.0** | **70.8** | **87.5** | **77.8** | **100** | **78.0** |
| LLaVA-1.6 | Raw | 20.0 | 20.8 | 0.0 | 11.1 | 50.0 | 18.0 |
| | **Ours** | **80.0** | **75.0** | **87.5** | **66.7** | **100** | **78.0** |
| InternVL-1.5 | Raw | 60.0 | 33.3 | 37.5 | 55.6 | 25.0 | 40.0 |
| | **Ours** | **80.0** | **83.3** | **87.5** | **77.8** | **100** | **84.0** |

## A.3 Experiments on UDK-VQA-20240905

We compare with state-of-the-art LVLMs on the UDK-VQA-20240905 test set, including Gemini 1.5 Pro [20], GPT-4o, LLaVA-1.6 [23] and InternVL-1.5 [45]. Similar to the principle for testing on the UDK-VQA dataset, we implement the ***Raw*** version of Gemini 1.5 Pro and GPT-4o via their official APIs, which do not have the ability of internet-augmented generation (IAG). For GPT-4o, we prompted its official web versions with "*Retrieve relevant news and answer the question directly from the given options using the option letters based on the image.*" to test its self-contained IAG-capable ability, dubbed ***IAG***. For LLaVA-1.6 and InternVL-1.5, we implement them via the VLMEvalKit toolkit [36]. Each Raw baseline is incorporated into our framework, dubbed as ***Ours***.

The experimental results on the UDK-VQA-20240905 test set are listed in Table 6, from which we can observe that: (1) InternVL-1.5 with our framework performs the best, suppressing closed-source business models including Gemini 1.5 Pro and GPT-4o. (2) All baseline models (Raw version) perform worse on the UDK-VQA-20240905 test set compared to the UDK-VQA test set, indicating that the newer the knowledge required by the test samples, the more challenging these samples are for LVLMs. (3) Compared to the experimental results on the UDK-VQA dataset, the improvement of our framework over the baseline has decreased. This is because there is a greater discrepancy in entity distribution between the UDK-VQA-0905 dataset and the training set of SearchLVLMs. (4) The accuracy of the Raw version of Gemini 1.5 Pro is lower, because this model tends to choose "E. No Correct Answers" while other models are more inclined to select from $\{A, B, C, D\}$, even though they do not possess the knowledge required to answer such questions correctly.

