# OpenReview forum: "SearchLVLMs: A Plug-and-Play Framework for Augmenting Large Vision-Language Models by Searching Up-to-Date Internet Knowledge"
_NeurIPS.cc/2024/Conference — NeurIPS 2024 poster_

### Official Review · Reviewer_PaHW · 2024-07-13

**Soundness:** 3
**Presentation:** 3
**Contribution:** 3
**Rating:** 6
**Confidence:** 1

**Summary:**

The paper presents UDKAG, a novel framework designed to enhance the capabilities of Large Vision-Language Models (LVLMs) by integrating up-to-date knowledge retrieved from the internet during the inference phase. The authors have developed a hierarchical filtering model to identify pertinent content from search engine results and a UDK-VQA dataset to evaluate the model's performance on VQA tasks.

**Strengths:**

1. The paper introduces an novel internet-augmented generation framework to update newest knowledge.
2. The creation of the UDK-VQA dataset is a valuable contribution.
3. Significant performance improvement.

**Weaknesses:**

I'm not familiar with this field and from my perspective this paper may not have obvious weakness.

**Questions:**

I'm curious about the generalization ability after integrating up-to-date knowledge. Will the model forget some important information and perform worse on other data?

**Limitations:**

Yes

---

> ### Author Rebuttal · Authors · 2024-08-07
>
> Thank you for your time and effort in reviewing our paper and for your recommended acceptance.
> We sincerely appreciate your valuable comments and feedback.
>
> ---
>
> > **Question: I'm curious about the generalization ability after integrating up-to-date knowledge. Will the model forget some important information and perform worse on other data?**
>
> **Answer:** The performance of large vision-language models (LVLMs) with our framework remains robust even on datasets that do not require up-to-date knowledge. We have conducted experiments on more existing datasets, including GQA [1], INFOSEEK [2] and A-OKVQA [3]. GQA is designed to evaluate the reasoning abilities of LVLMs and does Not Rely on external Knowledge **(NRK)**. INFOSEEK and A-OKVQA Require external Commonsense Knowledge **(RCK)**, such as "the Eiffel Tower is in Paris" and "the important function of a folding chair is portability", rather than up-to-date knowledge. In contrast, our UDKAG dataset Require Up-to-Date Knowledge **(RUDK)**. Our experimental results, presented in the table below, demonstrate that our framework consistently improves various baselines across these three datasets, underscoring the generalizability of our approach.
>
> ---
>
> | Baseline      | Variant   | Local Data (clear GT) | Internet Data (unclear GT) | GQA [1] (NRK) | INFOSEEK [2] (RCK) | A-OKVQA [3] (RCK) | UDKAG (RUDK) |
> |---------------|-----------|-----------------------|----------------------------|---------------|--------------------|-------------------|--------------|
> | CFR [5]       | -         | -                     | -                          | 72.10         | -                  | -                 | -            |
> | Oracle→FID [2]| -         | ✓                     | -                          | -             | 45.60              | -                 | -            |
> | Omni-SMoLA [4]| -         | ✓                     | -                          | -             | -                  | 84.10             | -            |
> | LLaVA-1.6     | Raw       | -                     | -                          | 61.66         | 37.86              | 75.53             | 31.8         |
> |               | **Ours**  | **-**                     | **✓**                          | **62.33**     | **41.25**          | **76.22**         | **90.2**     |
> | InternVL-1.5  | Raw       | -                     | -                          | 74.03         | 51.13              | 84.53             | 42.6         |
> |               | **Ours**  | **-**                    | **✓**                          | **74.41**     | **53.10**          | **84.59**         | **92.9**     |
>
> ---
> [1] Hudson et al. Gqa: A new dataset for real-world visual reasoning and compositional question answering. In CVPR 2019.
>
> [2] Chen et al. Can pre-trained vision and language models answer visual information-seeking questions? In EMNLP 2023.
>
> [3] Schwenk et al. A-okvqa: A benchmark for visual question answering using world knowledge. In ECCV 2022.
>
> [4] Wu et al. Omni-SMoLA: Boosting generalist multimodal models with soft mixture of low-rank experts. In CVPR 2024.
>
> [5] Nguyen et al. Coarse-to-fine reasoning for visual question answering. In CVPRW 2022.

---

> > ### Comment · Reviewer_PaHW · 2024-08-12
> >
> > I appreciate the author's commitment and valuable time in addressing my concerns. After reading through all the reviews and rebuttals, I believe this work deserves a broader discussion. I tend to maintain the 'weak accept' recommendation.

---

> > > ### Author Response · Authors · 2024-08-12
> > > **Thank you for your review!**
> > >
> > > Thank you for your prompt reply and valuable feedback on our manuscript. In the revised version, we will include experimental results on more datasets to better demonstrate the generalizability of our framework.

---

### Official Review · Reviewer_jo4p · 2024-07-13

**Soundness:** 4
**Presentation:** 4
**Contribution:** 3
**Rating:** 7
**Confidence:** 4

**Summary:**

This paper investigates a novel approach to augment large vision-language models with up-to-date knowledge from the internet. The author proposes to extensively leverage existing search engines (e.g., Bing and Google) and foundation models like ChatGPT for web information searching and parsing. A hierarchical filtering model has been proposed to filter the irrelevant information from the retrieved websites. The method can augment existing LVLM for augmented generations and achieve the state-of-the-art (SOTA) performance on a newly proposed VQA dataset, i.e., UDK-VQA, where the proposed plug-and-play method can improve SOTA LLM/VLMs like Gemini 1.5 Pro, GPT-4v, GPT-4o and LLaVA 1.6. Extensive ablation studies have been conducted to analyze the effectiveness of each component of the proposed framework.

**Strengths:**

1. The paper is very well written and strongly motivated, given the necessity of augmenting the LVLM with up-to-date knowledge. To the best of my knowledge, this should be the first paper on augmenting the LVLM with internet information in a comprehensive manner.

2. The proposed method is novel, given its design choice and intelligent combination of search engine and foundation model to enable precise information retrieval from the internet. The reviewer believes that such a new systematic system will inspire future exploration of IAG for multi-modality foundation models.

3. The proposed method can be plug-and-play into many existing foundation models without trial and error, demonstrating its universality and significance. Moreover, the foundation models augmented by the proposed method can be improved significantly, highlighting the effectiveness of the proposed method.

4. Extensive analysis has been conducted on the proposed method to verify further the effectiveness of the proposed framework.

**Weaknesses:**

Overall, this is a strong paper. The primary concern is the cost of leveraging the search engine and LLM like ChatGPT. The author should elaborate more on this aspect to show whether such an IAG is cost-efficient. Moreover, the reviewer is also curious about whether the code of the proposed framework will be open-source to serve as the cornerstone for future research.

**Questions:**

Please refer to the Weaknesses section for more details.

**Limitations:**

N/A.

---

> ### Author Rebuttal · Authors · 2024-08-07
>
> Thank you for recognizing the significance of my work and for your constructive comments, which will help enhance the quality of my paper.
>
> ---
>
> > **Weakness1: The primary concern is the cost of leveraging the search engine and LLM like ChatGPT. The author should elaborate more on this aspect to show whether such an IAG is cost-efficient.**
>
> **Response:** Thanks for your suggestions.
> Here is a cost calculation for using our framework for inference. The main costs involved are generating queries with ChatGPT, calling Bing Text Search, and calling Bing Visual Search.
> The cost of ChatGPT is `$1.50` per 1M input tokens and  `$3.00` per 1M output tokens.
> The cost of Bing Text Search and Bing Visual Search are `$0.025` and `$0.015` per search, respectively.
> For our framework, testing a VQA sample involves approximately 50/20 input/output tokens for ChatGPT, 3 Bing Text Searches and 1 Bing Visual Search.
> Therefore, the total cost for testing a VQA sample using our framework is approximately `$0.09`.
>
> ---
>
> > **Weakness2: Moreover, the reviewer is also curious about whether the code of the proposed framework will be open-source to serve as the cornerstone for future research.**
>
> **Response:** We will release the dataset and the code once the paper is accepted.

---

> ### Comment · Reviewer_jo4p · 2024-08-13
>
> Thanks the author for the detailed response! The analysis of the cost for the search engine further increase the significance of the paper. I tend to keep my score as "Accept".

---

> > ### Author Response · Authors · 2024-08-13
> > **Thank you for your review!**
> >
> > Thank you for your positive feedback and for noting the significance of the cost analysis in my paper. I appreciate your time and thoughtful evaluation.

---

### Official Review · Reviewer_qhJo · 2024-07-19

**Soundness:** 2
**Presentation:** 3
**Contribution:** 2
**Rating:** 4
**Confidence:** 4

**Summary:**

The paper proposes a framework that enhances large vision-language models (LVLMs) to handle visual question answering (VQA) tasks involving up-to-date knowledge. The system utilizes a search engine to retrieve relevant websites and parses their content for more comprehensive information. To manage the large volume of data, a hierarchical filtering model is introduced, consisting of a website filter and a content filter. These filters score and select the most relevant information, which is then used to prompt LVLMs for improved performance on VQA tasks. The framework also includes a diversity selection mechanism to ensure varied and unbiased content selection.

**Strengths:**

1. Clear Motivation: The paper is well-motivated and addresses a very practical problem.
2. Hierarchical Filtering Model: The two-step filtering process effectively manages large data volumes, enhancing the performance of LVLMs by ensuring that only the most relevant content is used.
3. Comprehensive Evaluation: The introduction of the UDK-VQA dataset and the experimental comparisons with LVLMs seem to to provide evidence of the framework’s effectiveness.

**Weaknesses:**

1. "First open-source framework which seamlessly incorporates existing LVLMs to equip them with up-to-date knowledge during inference." I am concerned about this statement. There have been many frameworks utilizing external knowledge. Many of them can seamlessly apply to LVLMs. I am wondering what makes the proposed method different from them.

2. Lack of Comparison against Existing SOTA Methods Utilizing External Knowledge: I encourage the author to conduct experiments to include more SOTA methods directly utilizing retrieved external knowledge for clear comparison.

2. The author mentions the construction of the dataset as one of the contributions. However, I am concerned there is limited novelty in the construction pipeline of UDK-VQA as the steps are very engineering and practical.

3. The paper claims UDK-VQA to be the first test set for evaluating the ability of LVLMs in handling VQA about up-to-date knowledge. I have concerns about this. There have been plenty of VQA benchmarks requiring external knowledge in the field. I am wondering what really makes UDK-VQA different from the existing ones. Are the up-to-date knowledge? However, this terminology is very blurry. What is the definition of up-to-date knowledge? How do you define the timestamp threshold?

4. Broader Evaluation on Other Benchmarks Requiring External Knowledge: Extend the evaluation to include more diverse datasets and LVLMs to verify the generalizability of the framework across different benchmarks requiring external knowledge.

5. Performance Dependency: The framework’s performance is heavily dependent on the quality of the hierarchical filtering model, which may vary across different LVLMs and datasets.

**Questions:**

Please see above.

**Limitations:**

The authors have made a commendable effort to address the limitations of their work, including 1) Separate Training for Filters and 2) LVLMs and   Content Snippet Limitations.

Suggestions for Improvement:
    1) Detailed Impact of Separate Training: The authors could provide more insights into how the separate training of filters and LVLMs specifically impacts performance and whether integrating these components could lead to substantial improvements.
    2) Addressing Incomplete Snippets: While the paper acknowledges the limitation of incomplete snippets, it would be beneficial to discuss potential strategies to enhance snippet completeness or alternative methods to initially extract more comprehensive information.

---

> ### Author Rebuttal · Authors · 2024-08-07
>
> Thank you for taking the time to review our paper.
> We value your suggestions.
> Here, we would like to mention that our work focuses on multimodal Internet-Augmented Generation (IAG), a type of method to enhance LVLMs in handling multimodal prompts based on Internet knowledge.
> IAG is an emerging topic in the field of Retrieval-Augmented Generation (RAG), offering significant potential and application value.
> For example, OpenAI's recently revealed SearchGPT employs IAG as one of its main technologies.
> We have carefully addressed your comments, but due to the length, **some responses have been placed in the comments section**.
>
> ---
> > **Weakness1:
> Differences from existing frameworks.**
>
> **Response:**
> Our proposed framework significantly differs from existing works. Existing frameworks for LVLMs utilizing external knowledge can be categorized into two types: (1) Frameworks that retrieve from local knowledge bases. (2) Frameworks based on external tools. The first type of framework [2][6] often retrieves external commonsense knowledge, such as "the Eiffel Tower is in Paris" and "the important function of a folding chair is portability". This type of knowledge is not frequently updated and can be easily embedded in LVLMs during training. Differently, we focus on the up-to-date knowledge, which is frequently updated over time, such as "how many gold medals did the USA win in the Olympics on August 6, 2024.". LVLMs cannot acquire this type of knowledge during the training phase. The second type of framework [7][8] uses search engines to obtain the up-to-date knowledge knowledge, but they focus on the orchestration of different tools. Differently, we focus on how to retrieve and use the obtained knowledge to augment LVLMs, which is orthogonal to the second type of framework and our method can be embedded into their framework.
>
> ---
> > **Weakness2:
> Lack of comparison against existing SOTA methods utilizing external knowledge.**
>
> **Response:**
> Thanks for your suggestions.
> We immediately conducted experiments using more methods: GENREAD [6], Chameleon [7], and CLIP→FID [2].
> GENREAD uses LLMs for knowledge retrieval and cannot access the up-to-date knowledge. Chameleon is an external tool-based framework that uses snippets returned by search engines for augmentation. CLIP→FID is a framework based on local database retrieval, and we integrated it with some components of our framework to allow it to access the up-to-date knowledge.
> The experimental results are shown in the table below, where **QG** denotes Query Generator and **HFM** denotes Hierarchical Filtering Model.
> We can observe that our framework achieves the best performance, proving the effectiveness of our approach.
> If you have any suggestions for frameworks to compare, please let us know. We look forward to discussing them with you.
>
> | Baseline   | Variant       | QG (*Q*) | QG (*V*) | HFM   | UDKAG  |
> |------------|---------------|----------|----------|-------|--------|
> | LLaVA 1.6 | Raw           | -        | -        | -     | 31.8   |
> |            | GENREAD [6]   | -        | -        | -     | 31.3   |
> |            | Chameleon [7] | -        | -        | -     | 62.3   |
> |            | CLIP→FID [2]  | ✓        | ✓        | -     | 54.7   |
> |            | **Ours**      | **✓**        | **✓**        | **✓**     | **90.2** |
> | InternVL 1.5 | Raw           | -        | -        | -     | 42.6   |
> |            | GENREAD [6]  | -        | -        | -     | 27.6   |
> |            | Chameleon [7] | -        | -        | -     | 59.3   |
> |            | CLIP→FID [2]  | ✓        | ✓        | -     | 57.7   |
> |            | **Ours**      | **✓**        | **✓**        | **✓**     | **92.9** |
> | Qwen-VL    | Raw           | -        | -        | -     | 35.2   |
> |            | GENREAD [6]   | -        | -        | -     | 23.5   |
> |            | Chameleon [7] | -        | -        | -     | 43.8   |
> |            | CLIP→FID [2]  | ✓        | ✓        | -     | 36.7   |
> |            | **Ours**      | **✓**        | **✓**        | **✓**     | **84.8** |
> | LLaVA 1.5  | Raw           | -        | -        | -     | 41.2   |
> |            | GENREAD [6]   | -        | -        | -     | 31.9   |
> |            | Chameleon [7] | -        | -        | -     | 60.7   |
> |            | CLIP→FID [2]  | ✓        | ✓        | -     | 58.5   |
> |            | **Ours**      | **✓**        | **✓**        | **✓**     | **88.9** |
>
> ---
> > **Weakness3:
> Limited novelty in the construction pipeline of UDK-VQA.**
>
> **Response:**
> (1) Automatically generating high-quality internet VQA samples is challenging.
> We employed several strategies to improve the quality of the generated data, resulting in VQA samples that can be used for training. The strategies include using an entity replacement strategy to avoid generating samples with meaningless images, and using consistency answering (as described in Section 4.2, second paragraph) and image clustering to automatically filter out incorrect samples.
> (2) With our pipeline, it is easy to collect a large number of VQA samples for training and regularly generate test data that is free from data contamination issues.

---

> ### Author Response · Authors · 2024-08-07
> **Response to Reviewer qhJo**
>
> > **Weakness4:
> Differences between UDK-VQA and existing datasets. Definition of up-to-date knowledge and timestamp threshold.**
>
> **Response:**
> The difference between our UDK-VQA dataset and existing datasets lies in the timeliness of the test data.
> Existing RAG datasets rely on external knowledge that is usually commonsense knowledge, which is not time-sensitive and does not update frequently. Such knowledge can be learned by LVLMs during pre-training and supervised fine-tuning.
> In contrast,
> the knowledge our dataset relies on is time-sensitive and belongs to the up-to-date knowledge, which refers to the most current and relevant information, data, and understanding available at a given point in time.
> Specifically, this refers to information that emerged after the LVLMs completed their training, such as news generated after a certain date.
> We set the timestamps for data collection to be after the latest release date of all LVLMs we tested, ensuring that the knowledge required by the test set was not accessible to LVLMs during their training.
> This work is a long-term project, and we will continue to update the test set automatically using our framework to ensure the data is new for LVLMs.
>
> > **Weakness5:
> Broader evaluation on other benchmarks requiring external knowledge.**
>
> **Response:**
> Thanks for your suggestions.
> We have evaluated different LVLMs on a broader benchmark, including GQA [1], INFOSEEK [2], A-OKVQA [3] and the proposed UDKAG.
> The experimental results are shown in the table below, where GQA does Not Rely on external Knowledge **(NRK)**, INFOSEEK and A-OKVQA Rely on Commonsense Knowledge **(RCK)**, and UDKAG Relies on the Up-to-Date Knowledge **(RUDK)**.
> From the table, we can observe that our framework improves the performance of different LVLMs across various datasets.
> The improvements on these three datasets are not as significant as on our UDKAG dataset for the following reasons:
> (1) GQA does not rely on external knowledge and is used to evaluate the reasoning ability of LVLMs, which is beyond the scope of our framework.
> (2) Our framework focuses on retrieving the up-to-date knowledge, whereas INFOSEEK and A-OKVQA rely on commonsense knowledge, much of which has already been used in the training data of LVLMs.
>
>
> | Baseline      | Variant   | Local Data (clear GT) | Internet Data (unclear GT) | GQA [1] (NRK) | INFOSEEK [2] (RCK) | A-OKVQA [3] (RCK) | UDKAG (RUDK) |
> |---------------|-----------|-----------------------|----------------------------|---------------|--------------------|-------------------|--------------|
> | CFR [5]       | -         | -                     | -                          | 72.10         | -                  | -                 | -            |
> | Oracle→FID [2]| -         | ✓                     | -                          | -             | 45.60              | -                 | -            |
> | Omni-SMoLA [4]| -         | ✓                     | -                          | -             | -                  | 84.10             | -            |
> | LLaVA-1.6     | Raw       | -                     | -                          | 61.66         | 37.86              | 75.53             | 31.8         |
> |               | **Ours**  | **-**                     | **✓**                          | **62.33**     | **41.25**          | **76.22**         | **90.2**     |
> | InternVL-1.5  | Raw       | -                     | -                          | 74.03         | 51.13              | 84.53             | 42.6         |
> |               | **Ours**  | **-**                    | **✓**                          | **74.41**     | **53.10**          | **84.59**         | **92.9**     |
>
> ---
>
> > **Weakness6:
> Performance dependency on the hierarchical filtering model.**
>
> **Response:**
> Our experimental results have proven that: (1) The hierarchical filtering model has transferability and can be applied to different LVLMs and datasets without fine-tuning. (2) The quality of the hierarchical filtering model is not significantly related to the backbone it uses. As shown in Table 1 and Figure 4 of the main manuscript, we tested our hierarchical filtering model on 13 different LVLMs, all of which showed significant improvement. As described in our response to weakness 5, our hierarchical filtering model can also bring some improvement when directly transferred to other datasets, whether they are based on additional knowledge or not. Furthermore, as shown in Table 2 of the main manuscript, using LLaVA and Qwen as the backbone for the hierarchical filtering model can achieve significant performance improvements.

---

> ### Author Response · Authors · 2024-08-07
> **Response to Reviewer qhJo**
>
> > **Limitation1:
> Detailed impact of separate training.**
>
> **Response:**
> Separate training is necessary as joint training results in performance degradation. As shown in the table below, whether using Qwen-VL or LLaVA-1.5 as the baseline model, separate training brings more significant performance improvement. The main reasons are: (1) Our training data uses pseudo-labeling instead of high-quality human annotation. Training based on such data may cause LVLMs to lose their original semantic understanding capabilities. (2) Our training and testing sets are generated from news from different time periods, involving different entities and having different distributions. Training LVLMs on our training set easily leads to overfitting, resulting in lower generalization on the test set.
>
> | Baseline  | Variant      | Training Strategy  | UDKAG |
> |-----------|--------------|--------------------|-------|
> | QWen-VL   | Raw          | -                  | 35.2  |
> |           | Ours         | Joint Training     | 68.5  |
> |           | **Ours**     | **Separate Training**  | **84.8** |
> | LLaVA 1.5 | Raw          | -                  | 41.2  |
> |           | Ours         | Joint Training     | 68.0  |
> |           | **Ours**     | **Separate Training**  | **88.9** |
>
> ---
>
> > **Limitation2:
> Addressing incomplete snippets.**
>
> **Response:**
> Thanks for your suggestions.
> The completeness of snippets has little impact on accuracy because their purpose is to allow the website filter to perform an initial screening of web pages, reducing the input the content filter needs to process, thereby improving the efficiency of our framework. Directly using incomplete snippets with existing search engines is already widely practiced [7][8]. There are two intuitive strategies to enhance the completeness of snippets: (1) Using large language models (LLMs) to continue writing the snippets. (2) Crawling the complete content of websites to fill in the snippets. The first method may result in incorrect snippets because LLMs lack the up-to-date knowledge and cannot accurately continue the writing. The second method will incur additional time consumption during inference because it requires crawling the content of all websites, regardless of whether the content filter needs to process that content. This contradicts our original intention of improving efficiency with the website filter. We are implementing the second method to verify whether complete snippets will bring performance improvements, and the experimental results will be provided in the discussion phase.
>
> ---
>
> [1] Hudson et al. Gqa: A new dataset for real-world visual reasoning and compositional question answering. In CVPR 2019.
>
> [2] Chen et al. Can pre-trained vision and language models answer visual information-seeking questions? In EMNLP 2023.
>
> [3] Schwenk et al. A-okvqa: A benchmark for visual question answering using world knowledge. In ECCV 2022.
>
> [4] Wu et al. Omni-SMoLA: Boosting generalist multimodal models with soft mixture of low-rank experts. In CVPR 2024.
>
> [5] Nguyen et al. Coarse-to-fine reasoning for visual question answering. In CVPRW 2022.
>
> [6] Yu et al. Generate rather than retrieve: Large language models are strong context generators. In ICLR 2023.
>
> [7] Lu et al. Chameleon: Plug-and-play compositional reasoning with large language models. In NeurIPS 2023.
>
> [8] Yang et al. Mm-react: Prompting chatgpt for multimodal reasoning and action. In arXiv 2023.

---

> ### Author Response · Authors · 2024-08-11
> **Message to Reviewer qhJo**
>
> Dear Reviewer qhJo,
>
> We hope this message finds you well.
>
> We sincerely appreciate the time and effort you are dedicating to evaluating our submission. As the conference timeline approaches important deadlines, we would be grateful for any updates you could provide regarding the review status of our paper. If there is anything further we can provide to assist with the review process, please don’t hesitate to let us know.
>
> Additionally, we present the **experimental results of snippet completeness** on performance here, where $\theta$ represents the percentage as mentioned in Section 5.6 of the main manuscript. For each website snippet, we attempted to locate the full sentence corresponding to the snippet by crawling the website's content. However, the content of many websites could not be crawled. For such websites, we experimented with two strategies: (1) Discarding these websites during training and testing. (2) Using the incomplete snippets. The experimental results are shown in the table below, where "Raw" represents all snippets without completion, "Discard" represents strategy (1), and "Mixture" represents strategy (2). The experimental results show that directly discarding the websites leads to a significant performance loss, as discarding reduces the number of usable websites by approximately half, thereby limiting the performance of the website filter. As $\theta$ increases, the performance of strategy (2) becomes increasingly close to that of all snippets without completion, which validates our rebuttal argument that the completeness of the snippets has little impact on accuracy.
>
> We greatly appreciate your attention to this matter and look forward to any feedback you may have.
>
> Best regards,
>
> Authors
>
> ---
> | Baseline  | Variant  | $\theta$=10\% | $\theta$=25\% | $\theta$=40\% | $\theta$=55\% | $\theta$=70\% | $\theta$=85\% | $\theta$=100\% |
> |-----------|----------|-------------|-------------|-------------|-------------|-------------|-------------|--------------|
> | LLaVA-1.6 | Raw      | 80.8        | 86.1        | 88.4        | 89.4        | 89.7        | 90.0        | 90.2         |
> |           | Discard  | 76.6        | 80.2        | 80.7        | 82.1        | 81.9        | 81.3        | 81.6         |
> |           | Mixture  | 83.6        | 87.9        | 89.4        | 89.6        | 89.8        | 90.4        | 90.2         |
> | InternVL-1.5 | Raw   | 84.6        | 89.2        | 91.1        | 92.9        | 92.4        | 92.7        | 92.9         |
> |           | Discard  | 79.7        | 82.2        | 82.9        | 83.8        | 84.0        | 82.8        | 83.2         |
> |           | Mixture  | 86.0        | 89.5        | 92.2        | 92.3        | 92.2        | 92.5        | 92.9         |

---

> ### Author Response · Authors · 2024-08-12
> **Hoping for further discussion with you.**
>
> Dear Reviewer qhJo,
>
> We express gratitude for your time spent on reviewing and your valuable comments. We have addressed your concerns by providing relevant responses and results. We look forward to engaging in further discussion to confirm whether or not your concerns have been addressed.
>
> Best regards,
>
> Authors

---

> ### Author Response · Authors · 2024-08-13
> **Looking Forward to Your Reply.**
>
> Dear Reviewer qhJo,
>
> I hope this message finds you well. We are truly grateful for the effort you have put into reviewing our manuscript and for the insightful feedback you have provided.
>
> We understand that you may have a busy schedule, but we kindly ask if you could review our responses at your earliest convenience. Your feedback is essential for us to move forward, and we would greatly value any additional insights you may have.
>
> Thank you once again for your time and consideration.
>
> Best regards,
>
> Authors

---

### Author Rebuttal · Authors · 2024-08-07

### **General response**

We would like to thank the Area Chairs and the Reviewers for carefully reading our paper and providing valuable comments.

We are pleased to hear that the majority of the reviewers found our paper "well-written " (jo4p) and "strongly motivated" (qhJo, jo4p).
We also appreciate the reviewers' recognition of the "novelty" (jo4p, PaHW) and the "universality and significance" (jo4p) of our work.
Additionally, we are glad to receive positive feedback on our experiments, which "provide evidence of the framework's effectiveness" (qhJo), "verify further the effectiveness of the proposed framework" (jo4p), and show "significant performance improvement" (PaHW).

We look forward to the upcoming discussion sessions and to making further contributions to the vision-and-language community.

---

### Decision · Program_Chairs · 2024-09-25

**Decision:**

Accept (poster)

**Comment:**

This paper receives mostly positive rates (1 accept, 1 weak accept, and 1 borderline reject) thanks to a novel internet-augmented generation framework with convincing experimental results. Although overall reviews are positive, reviews still have some concerns such as computational costs and incomplete snippets, which should be addressed in the final version.